# SelfEval: Leveraging the discriminative nature of generative models for evaluation

## Abstract

In this work, we show that text-to-image generative models can be 'inverted' to assess their own text-image understanding capabilities in a completely automated manner. Our method, called SelfEval, uses the generative model to compute the likelihood of real images given text prompts, making the generative model directly applicable to discriminative tasks. Using SelfEval, we repurpose standard datasets created for evaluating multimodal text-image discriminative models to evaluate generative models in a fine-grained manner: assessing their performance on attribute binding, color recognition, counting, shape recognition, spatial understanding. To the best of our knowledge SelfEval is the first automated metric to show a high degree of agreement for measuring text-faithfulness with the gold-standard human evaluations across multiple models and benchmarks. Moreover, SelfEval enables us to evaluate generative models on challenging tasks such as Winoground image-score where they demonstrate competitive performance to discriminative models. We also show severe drawbacks of standard automated metrics such as CLIP-score to measure text faithfulness on benchmarks such as DrawBench, and how SelfEval sidesteps these issues. We hope SelfEval enables easy and reliable automated evaluation for diffusion models.

## 1 Introduction

In the past few years, generative image models have rapidly advanced and state-of-the-art text-to-image models now generate high quality realistic images. While a lot of research effort is focussed on improving these models, their evaluation has received considerably less attention. Evaluations for text-to-image models typically focus on two aspects: (1) quality of the generated image; and (2) the alignment between the generated image and the input text, *i.e.*, the 'faithfulness' of the generation. The gold standard for evaluating text-to-image models is to compare generations from pairs of models using human judgement. However, using pairwise human evaluations does not scale to lots of models or generations, and it is an open question on how to convert them to ordinal metrics to rank models. Thus, automatic evaluations are commonly used as a proxy for comparing models.

In this work, we focus on automatic evaluations that measure the 'text faithfulness' of the generated image to the input text prompt. While automated evaluations for diffusion models are common, they typically rely on an external discriminative model, *e.g.*, CLIP to measure the 'relatedness' of the generated image to the input text. Instead, we ask the question can the diffusion model itself be used to measure the relatedness of an image-text pair and thus evaluate its own generations?

Most work using text-to-image diffusion models focusses on sampling good images from them given a text prompt. However, as shown in Figure 1, diffusion models can be used to estimate the conditional likelihood of an image $\mathbf{x}$ given a text prompt $\mathbf{c}$, *i.e.*, $p(\mathbf{x}|\mathbf{c})$. We propose SelfEval which is a practical way to estimate such likelihoods accounting for numerical issues arising in standard diffusion models. We show that these likelihoods can be used directly to evaluate the model's text-faithfulness. SelfEval repurposes standard multimodal image-text datasets such as Visual Genome, COCO and CLEVR to measure the model's text understanding capabilities. Our evaluation allows us to assess fine-grained aspects such as the model's ability to recognize colors, count objects *etc*. We apply our method to a wide variety of diffusion models: different types of image representations (pixel based, latent space based), different text encoders and different model sizes. SelfEval's automatic evaluation results are in agreement with the 'gold-standard' human judgements making SelfEval suitable for evaluation.

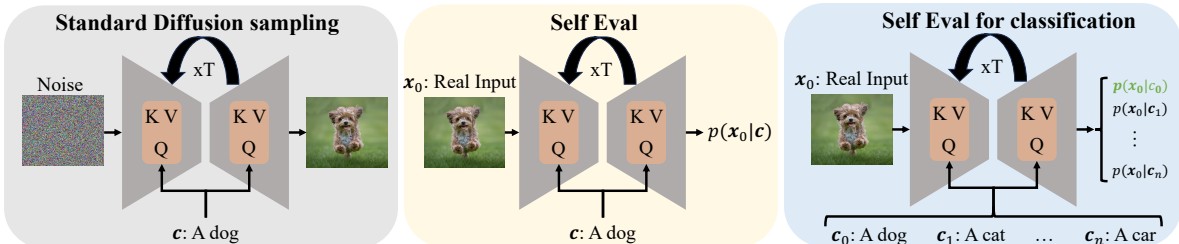

**Figure 1: Illustration of proposed method:** (Left) Starting from a noised input, the standard diffusion sampling method denoises the input iteratively to generate images from the input distribution. (Middle): SELFEVAL takes an image $\mathbf{x}_0$ and conditioning $\mathbf{c}$ pairs to estimates the likelihood $p(\mathbf{x}_0|\mathbf{c})$ of the pair in an iterative fashion. (Right): Given an image, $\mathbf{x}_0$ and $n$ captions, $\{\mathbf{c}_0, \mathbf{c}_1, \ldots, \mathbf{c}_n\}$, SELFEVAL is a principled way to convert generative models into discriminative models. In this work, we show that the classification performance of these classifiers can be used to evaluate the generative capabilities.

SELFEVAL has the added benefit that it does not require additional pretrained models apart from the diffusion models being compared. As we show in Figure 2, relying on an external model leads to three major issues. First, the automated metrics vary greatly depending on the type of the external model used for evaluation. Second, many generative models rely on an external model such as CLIP's text encoding during training, and thus using the same CLIP model for automated evaluation biases the results. Finally, the external model itself can be bad at certain image-text tasks, such as counting or attribute recognition making its scores unreliable for evaluations.

## 2  RELATED WORKS

**Generative models**: Generative models learn to model the joint distribution, $p(X, Y)$ of data consisting of an observed variable $X$ and the target $Y$. The model can subsequently be employed to generate novel data through sampling from the learned distribution.. In this work, we are interested in image generation models *i.e.*, models that learn the distribution of natural images. Generative Adverserial Networks (GAN) Goodfellow et al. (2014); Radford et al. (2015), Variational AutoEncoders (VAE) Kingma & Welling (2014) and Denoising Diffusion Probabilistic models (DDPM) Ho et al. (2020) are some of the most popular image generation models in the literature. GANs belong to the category of generative models, where two distinct components, a generator and a discriminator, are pitted against each other within a zero-sum game framework. VAEs are a category of autoencoders that ensure "regularity" within the latent space by constraining their distribution to closely align with a well-behaved and typically standard normal distribution. Subsequently, VQ-VAE's van den Oord et al. (2017) were proposed to prevent "posterior" collapse that were typically observed with VAEs. In more recent times, DDPMs have exceeded the capabilities of all preceding state-of-the-art image generative models in terms of their generative prowess. Drawing inspiration from non-equilibrium statistical physics, Diffusion probabilistic models Sohl-Dickstein et al. (2015) employ a forward diffusion process to gradually destroy the structure in the unknown input distribution and transforming it into a well-behaved and tractable distribution. A reverse diffusion process is trained to learn to restore the structure, thereby learning the input distribution. Ho et al. (2020) establish an explicit connection between diffusion models and denoising score matching Song & Ermon (2019); Vincent (2011) leading to a simplified objective for training diffusion models. In this study, we employ diffusion models owing to their exceptional image generation performance Dhariwal & Nichol (2021).

**Diffusion models**: In a relatively short time, diffusion models have surpassed GANs and VAEs as the defacto models for image generation due to their superior quality Dhariwal & Nichol (2021) and flexibility. Numerous studies have shown that diffusion models can be conditioned on a variety of modalities, including object classes Peebles & Xie (2023); Ho et al. (2020), natural language captions Saharia et al. (2022); Rombach et al. (2022); Nichol et al. (2022); Ramesh et al. (2022), camera pose Liu et al. (2023), images Brooks et al. (2023), bounding boxes Li et al. (2023b), segmentation, edge and depth maps Zhang & Agrawala (2023). Among these, text-conditioned diffusion models have attracted significant interest and popularity. Given paired image, caption data, the standard way of training text conditioned diffusion models is to fuse the caption features, extracted using a pre-trained text encoder, with the image features while training the reverse diffusion process. The fusion is typically done using cross-attentions Vaswani et al. (2017) layers. Models trained in this manner have demonstrated a remarkable comprehension of compositionality within text, often highlighted by their capac-

**Figure 2: Drawbacks of CLIP for generative model evaluation**. (Left) We compare the CLIP similarity scores of two Latent diffusion models Rombach et al. (2022) trained with CLIP ViT-L/14 (LDM-CLIP (ViT-L/14)) and OpenCLIP ViT-H/14 (LDM-CLIP (ViT-H/14)) text encoders. On the left, we compare the CLIP similarity scores, computed using CLIP ViT-L/14, on prompts generated from DrawBench, Winoground and, COCO datasets. The plot on the right compares the CLIP similarity scores computed using OpenCLIP ViT-H/14 model. The ranking changes depending on the model used. (Right) CLIP has poor performance in tasks involving counting instances, spatial relationships, matching attributes to objects and understanding corruption of text which constitute about 50 (25%) prompts in DrawBench. In each example, the correct caption is shown in green and CLIP picked the caption in bold. Using CLIP to evaluate text to image models on such prompts is not optimal.

ity to generate images based on counterfactual textual descriptions (like avacado shaped chair *etc.*). The most popular text encoders in use today for text-conditioned image synthesis are text encoders from the CLIP Radford et al. (2021) and the text-to-text transformer T5 Raffel et al. (2020). In this work, we analyze the text understanding capabilities of the diffusion models trained with different text encoders.

There exist two families of diffusion models in the literature, namely, pixel Saharia et al. (2022); Ramesh et al. (2022) and latent diffusion Rombach et al. (2022), differing primarily in the nature of input. As the name suggests, in pixel diffusion, the forward and reverse diffusion processes are performed on the pixels of the input. Performing diffusion on pixels is computationally expensive and hence a typical pixel diffusion pipeline consists of a low resolution generation step followed by a pixel upsampler to get a higher resolution output. Latent diffusion models Rombach et al. (2022) enable training with limited computational budget while simultaneously retaining their quality and flexibility. This is achieved by performing the diffusion process on the latent space of an autoencoder instead of the pixels. In this work, we analyze the text understanding capabilities of two state-of-the-art models with different text encoders each from pixel and latent diffusion models.

**Classifiers with diffusion models**: Lately, there has been a increase in the usage of conditional diffusion models as classifiers, driven by their superior understanding of the conditioned modality. These models are surprisingly good at capturing intricate patterns and dependencies within the conditioning input, making them strong discriminative models across a range of downstream tasks. Notable works include He et al. (2023), Mukhopadhyay et al. (2023) that either finetune a diffusion model, or use linear probing, for several classification and reasoning tasks. Unlike these methods, we do not train any models but instead convert the generative model into a discriminative one to understand its text understanding capabilities. Along similar lines as ours is the work of Clark & Jaini (2023) that observed that the zero-shot performance of the text-to-image pixel diffusion model, Imagen Saharia et al. (2022), is on-par with CLIP Radford et al. (2021) in several downstream tasks. Similar to Clark & Jaini (2023), Li et al. (2023a) adopt the standard ELBO loss as a proxy for the likelihood of a label, given an image, and convert it to a posterior probability using Bayes rule. Authors demonstrate impressive performance on several image classssification benchmarks. They also report promising results on ITM tasks on the Winoground Thrush et al. (2022) dataset. We propose a systematic way of estimating the likelihood scores from diffusion models and observe that the performance of generative classifiers, on several Image-Text Matching (ITM) tasks, can be used to evaluate their generative performance. To the best of our knowledge, we are the first method to compare the generative capabilities of different diffusion models using their discriminative performance.

## 3 METHOD: CONVERTING GENERATIVE MODELS TO DISCRIMINATIVE MODELS

Our method converts generative (diffusion) models into discriminative models by simply changing the inference, and does not require any retraining. This allows us to use the diffusion model itself on a variety of different image-text benchmarks and assess the diffusion model's image-text understanding capabilities. We briefly discuss an overview of diffusion models in Sec. 3.1 followed by our proposed method in Sec. 3.2

### 3.1 PRELIMINARIES

Diffusion Probabilistic Models (DPM) Sohl-Dickstein et al. (2015) are a class of generative models that are trained to 'denoise' inputs constructed by a Markovian *forward* process. The forward process starts with a real sample $\mathbf{x}_0$ and repeatedly adds gaussian noise, over $t$ timesteps to generate $\mathbf{x}_t$:

$$q(\mathbf{x}_t|\mathbf{x}_{t-1}) \sim \mathcal{N}(\mathbf{x}_t; \sqrt{1-\beta_t}\mathbf{x}_{t-1}, \beta_t\mathbf{I}). \tag{1}$$

Here $q(\mathbf{x}_0)$ is the data distribution. $\beta_t$ is the strength of the noise at timestep $t$ with $\beta_0 = 0, \beta_T = 1$. Note that $\mathbf{x}_t$ are the same size as the input. The joint distribution of the input along with the latents $q(\mathbf{x}_{0:T})$ is

$$q(\mathbf{x}_{0:T}) = q(\mathbf{x}_0) \prod_{t=1}^{T} q(\mathbf{x}_t|\mathbf{x}_{t-1}) \tag{2}$$

To sample images, one applies the *reverse* process $p(\mathbf{x}_{t-1}|\mathbf{x}_t)$, starting with $\mathbf{x}_T$ sampled from the unit normal distribution $\mathcal{N}(\mathbf{0}, \mathbb{I})$. So the joint distribution of the reverse process can be described as

$$p(\mathbf{x}_{0:T}) = p(\mathbf{x}_T) \prod_{t=1}^{T} p(\mathbf{x}_{t-1}|\mathbf{x}_t) \tag{3}$$

The reverse process $p(\mathbf{x}_{t-1}|\mathbf{x}_t)$ is not tractable and is often modeled using a neural network whose parameters are characterized by $\theta$, *i.e.* $p_\theta(\mathbf{x}_{t-1}|\mathbf{x}_t) \sim \mathcal{N}(\mathbf{x}_{t-1}; \boldsymbol{\mu}_\theta(\mathbf{x}_t, t), \boldsymbol{\Sigma}_\theta(\mathbf{x}_t, t))$.

### 3.2 LIKELIHOOD ESTIMATES FROM DIFFUSION MODELS

We specifically focus on text-to-image diffusion models, although our formulation extends to any conditional diffusion model. Text-to-image diffusion models are trained on a large datasets of image-text $(\mathbf{x}, \mathbf{c})$ pairs and model the reverse diffusion process $p(\mathbf{x}_{t-1}|\mathbf{x}_t, \mathbf{c})$. We 'invert' such a generative model and use it to estimate the likelihood of a real image $\mathbf{x}$ given a text caption $\mathbf{c}$, *i.e.*, $p(\mathbf{x}|\mathbf{c})$. We note that our method only changes the inference of a diffusion model and does not require any training. Assuming uniform prior on the classes, the likelihood $p(\mathbf{x}|\mathbf{c})$ can be converted into the posterior, $p(\mathbf{c}|\mathbf{x})$ using Bayes' rule, *i.e.* $p(\mathbf{c}|\mathbf{x}) = \frac{p(\mathbf{x}|\mathbf{c})}{|\mathcal{C}|}$, where $\mathcal{C}$ is the set of all classes.

Given the reverse process of a diffusion model parameterized by $\theta$, the likelihood for a datapoint $\mathbf{x}_0$ is

$$p_\theta(\mathbf{x}_0|\mathbf{c}) = \int p_\theta(\mathbf{x}_{0:T}|\mathbf{c})d\mathbf{x}_{1:T} \tag{4}$$

$$= \int p(\mathbf{x}_T) \prod_{t=1}^{T} p_\theta(\mathbf{x}_{t-1}|\mathbf{x}_T, \mathbf{c})d\mathbf{x}_{1:T}. \tag{5}$$

Since the diffusion models reverse process $p_\theta(\cdot)$ is also a gaussian, we can further write this as

$$p(\mathbf{x}_0|\mathbf{c}) = \int p(\mathbf{x}_T) \prod_{t=1}^{T} \frac{1}{\sqrt{(2\pi)^D|\Sigma_\theta|}} \exp(\frac{-(\mathbf{x}_{t-1} - \boldsymbol{\mu}_\theta(\mathbf{x}_t, \mathbf{c}))^T\boldsymbol{\Sigma}_\theta^{-1}(\mathbf{x}_{t-1} - \boldsymbol{\mu}_\theta(\mathbf{x}_t, \mathbf{c}))}{2})d\mathbf{x}_{1:T} \tag{6}$$

Here, $p(\mathbf{x}_T) \sim \mathcal{N}(0, \mathbb{I})$. For the sake of simplicity, we denote any realization of the random variable $\mathbf{x}_0$ as $\mathbf{x}_0$. Given a natural language caption $\mathbf{c}$, an image $\mathbf{x}_0$ and the noised latents $x_{1:T}$, the quantity inside the integral in Eq. 6 can be estimated numerically. We compute a Monte Carlo estimate of the integral by sampling $N$ noise terms ($\epsilon$) and computing $p(\mathbf{x}_0|\mathbf{c})$ as

$$p(\mathbf{x}_0|\mathbf{c}) = \sum_{n=1}^{N} p(\mathbf{x}_T^n) \prod_{t=1}^{T} p(\mathbf{x}_{t-1}^n|\mathbf{x}_t^n, \mathbf{c}) \quad \text{where } \mathbf{x}_t^n = \sqrt{1-\beta_t}\mathbf{x}_{t-1}^n + \sqrt{\beta_t}\epsilon^n \tag{7}$$

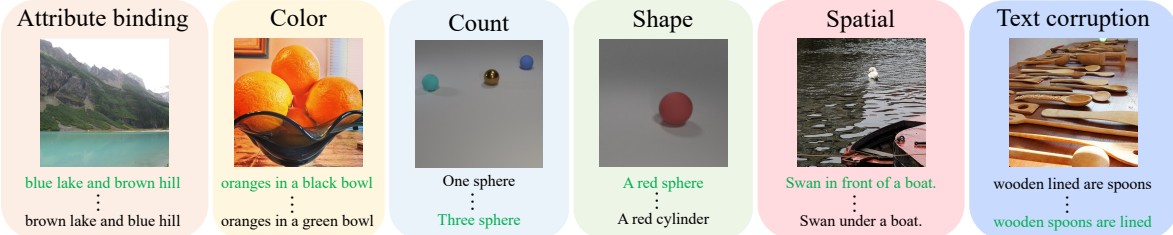

**Figure 3: Representative samples from the benchmark.** We divide the evaluation into six broad tasks, namely `Attribute binding`, `Color`, `Count`, `Shape`, `Spatial`, and `Text Corruption`. Each task is designed to evaluate a specific aspect of text faithfulness mimicking the categories in DrawBench. Each task is posed as an image-text matching problem, where given an image, the goal is to pick the right caption among distractors. The figure above shows examples from each task with the right caption highlighted in green.

**Practical considerations.** The terms on the RHS of Eq. 7 are multivariate gaussians and analytically computing them involves exponentials which can be numerically unstable. This can be prevented by computing log probabilities instead. Taking log both sides of Eq. 7, we get

$$\log p(\mathbf{x}_0|\mathbf{c}) = \log \sum_{n=1}^{N} p(\mathbf{x}_T^n) \prod_{t=1}^{T} p(\mathbf{x}_{t-1}^n|\mathbf{x}_t^n, \mathbf{c}) \tag{8}$$

$$\geq \sum_{n=1}^{N} \left( \log p(\mathbf{x}_T^n) + \sum_{t=1}^{T} \log p(\mathbf{x}_{t-1}^n|\mathbf{x}_t^n, \mathbf{c}) \right) \tag{9}$$

Where the last inequality is due to Jensen's inequality for concave functions, *i.e.* $\mathbb{E}(f(x)) \leq f(\mathbb{E}(x))$. All the terms in Eq. 9 are log probabilities of multivariate gaussians, which can be computed analytically and are numerically more stable.

We now show how estimating $p(\mathbf{x}_0|\mathbf{c})$ allows us to use a diffusion model for discriminative tasks and thus to evaluate their image-text understanding capabilities.

### 3.3 SELFEVAL TO EVALUATE DIFFUSION MODEL'S TEXT FAITHFULNESS

The 'standard' way of evaluating the text faithfulness of generative models is to 1. manually construct a prompt set that can reason about the model's capacity to understand the input text, and, 2. evaluate the faithfulness of the generation using human evaluators or an automatic evaluation score, like the CLIP Radford et al. (2021) score. As evaluating the text faithfulness of generative models inherently involves vision-language reasoning, we propose an alternate way to evaluate the model's performance across discriminative image-text reasoning tasks. In SELFEVAL, we pose the evaluation as an image-text matching problem and repurpose standard discriminative image-text datasets. Thus, SELFEVAL does not rely on external models such as CLIP, does not need human evaluators, and does not need manual prompt-set curation.

Image-text matching problems such as image-classification or retrieval can be reformulated as picking the correct caption for a single image $\mathbf{x}$ from a set of captions $\{\mathbf{c}_i\}$. We can use a diffusion model to estimate $p(\mathbf{x}|\mathbf{c}_i)$ for each of the captions and pick the caption that gives the highest likelihood. As shown in Fig. 1, the noised latents $\mathbf{x}_{1:T}$ are computed using the forward process. The final latent $\mathbf{x}_T$ is denoised for $T$ steps using the reverse process to obtain the denoised latents $\bar{\mathbf{x}}_{0:T-1}$. This process is repeated for $N$ independent noise vectors resulting in $\{\mathbf{x}_{1:T}^n\}_{n=1}^{N}$, $\{\bar{\mathbf{x}}_{0:T-1}^n\}_{n=1}^{N}$. Next, the likelihood can be computed as $p(\mathbf{x}_0|c_k) = \sum_{n=1}^{N} p(\mathbf{x}_T^n) \prod_{t=1}^{T} p(\mathbf{x}_{t-1}^n|\bar{\mathbf{x}}_t^n, \mathbf{c}_k)$, which is then converted to the posterior, $p(\mathbf{c}_k|\mathbf{x}_0)$ using Bayes' rule. Finally, the caption with the highest likelihood, *i.e.* $\arg\max_{\mathbf{c}_i} p(\mathbf{c}_i|\mathbf{x}_0)$ is chosen as the right one.

## 4 EXPERIMENTS

We now use SELFEVAL to evaluate text-to-image diffusion models. In § 4.1, we introduce our benchmark datasets and models, and present the SELFEVAL results in § 4.2.

### 4.1 BENCHMARK AND EVALUATION

In SELFEVAL, we pose the text faithfulness evaluation as an image-text matching problem, where the goal is to pick the right image caption pair among distractors.

**Tasks.** We identify and divide the evaluation into six broad reasoning tasks (illustrated in Figure 3): 1) Attribute binding, 2) Color, 3) Count, 4) Shape, 5) Spatial relationships, and 6) Text corruption. Each of these tasks evaluate the model's understanding of a specific aspect of text faithfulness and is similar to the categories of prompts from DrawBench Saharia et al. (2022). The six tasks are constructed using data from TIFA Hu et al. (2023), CLEVR Johnson et al. (2016) and ARO Yuksekgonul et al. (2023).

**Datasets. TIFA** Hu et al. (2023) consists of 4000 text prompts, collected manually and from image captioning datasets, to evaluate the text faithfulness of generative models. In our evaluation, we use ∼2000 of these text-prompts that are constructed from the COCO Lin et al. (2014) dataset and convert the dataset from question-answering to an image-text matching format as detailed in the supplement. **Attribution, Relation and Order (ARO)** Yuksekgonul et al. (2023) is a benchmark that uses data from Visual Genome Krishna et al. (2017) for attribute and spatial relations, and COCO for ordering tasks. **CLEVR** Johnson et al. (2016) is a benchmark for compositional understanding and visual reasoning that uses synthetic images. We use the splits proposed by Lewis et al. (2022) for our experiments.

We divide the datasets among all the reasoning task as follows. For attribute binding, we combine samples from ARO (attribution) and CLEVR. For colors and counts, we use corresponding samples from TIFA and CLEVR. For shapes, use samples from CLEVR. Data for spatial relationships is from TIFA, CLEVR and ARO (relations). The data for the text corruption task is from the ARO (order sensitivity) dataset. A sample of each task consists of an image and multiple text prompts and the performance on the task is the classification accuracy of pairing the image with the right caption.

We measure the performance of text-to-image generative models on the benchmark using the following evaluation methods. We provide full details for each of the methods in the supplement.

**SELFEVAL (Ours)** is an automatic evaluation method and uses both the images and text from our benchmark introduced in § 4.1. For each benchmark task, we randomly sample 1000 examples and evaluate the classification performance on them. We repeat this three times and the report the mean accuracy. We use 10 trials (*i.e.* $N = 10$) and perform diffusion for 100 steps (*i.e.* $T = 100$) for all the models. Refer to the supplement for ablation experiments on $N, T$.

**Human evaluations** are the gold standard for judging the performance of text-to-image models using pairwise comparsions. We present humans with generations from two models and ask them to vote for one of four choices: "both" the generations are faithful, "none" of them are faithful, or if only one of the two images ("Image 1" or "Image 2") demonstrates fidelity to the given prompt. For simplicity, we only report votes where there is a clear preference for a model. We randomly pick 250 text prompts from each benchmark task as conditioning for human evaluation and the images are generated using DDIM Song et al. (2021) sampling, with 100 denoising steps. Note that unlike SELFEVAL, human evaluations do *not* use the real images from the benchmark tasks and the human evaluators only look at the generated images.

#### 4.1.1 MODELS

We use models with different image representations: pixel diffusion models which directly use the pixel RGB values, and latent diffusion models where the image is projected into a latent space using an auto-encoder. We pick models trained with different text encoders within each class. This enables us to analyze the effect of text encoder on the final performance within each class.

**Diffusion models with CLIP text encoder.** For latent diffusion, we use a model trained with OpenCLIP Ilharco et al. (2021) text encoder with a ViT-H/14 backbone via an API containing open-sourced model weights. This model is trained on a public dataset with 5 billion images, excluding explicit material, and outputs $512 \times 512$

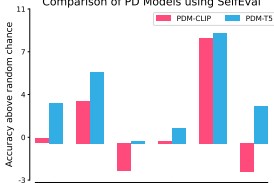 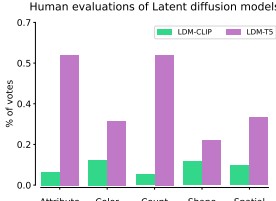 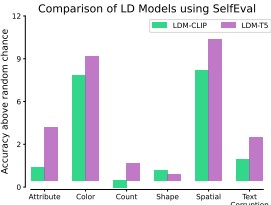

**Figure 4: Evaluating text-to-image models** using human evaluations and SELFEVAL. We evaluate different types of text-to-image models such as pixel diffusion (first two columns) and latent diffusion model (last two columns), and models that use different text encoders such as T5 XXL and CLIP. We observe that across all 4 diffusion models the relative ordering given by SELFEVAL's accuracy correlates with the pairwise human evaluation results. We also observe that latent diffusion models have a higher SELFEVAL accuracy than pixel diffusion models suggesting better text-faithfulness. Using the stronger T5 text encoder leads to better performance across human evaluations and SELFEVAL.

images. For pixel diffusion, we adopt the architecture of DALLE-2 Ramesh et al. (2022) for our experiments and train a model. We use a CLIP (ViT-L/14) text encoder and produce images of resolution $64 \times 64$. Our model has a total of 4.2B parameters and is trained for 2M steps on an internal image-text dataset (Internal-Dataset).

**Diffusion models with T5 text encoder.** For latent diffusion, we train a UNet model, similar to Rombach et al. (2022), but replace the CLIP text encoder with a T5 XXL Raffel et al. (2020) text encoder that outputs images of resolution $256 \times 256$. This model is also trained on Internal-Dataset for 2M steps using a latent space with a $4\times$ downsampling factor and has a total of 5.8B parameters. We train a 7.5B parameter pixel diffusion model, similar to Imagen Saharia et al. (2022), on inputs of resolution $64 \times 64$ for 2M steps also on Internal-Dataset. Subsequently, we apply a super resolution model to upsample the output to $512 \times 512$.

With the exception of the CLIP-based latent diffusion model Rombach et al. (2022), all the other models are trained for the same number of steps on the exact same data to ensure fair comparison.

## 4.2 MAIN RESULTS

We evaluate the four text-to-image models and report results in Figure 4. For SELFEVAL, we report the accuracy difference with the random chance accuracy, since each of the tasks has a different degree of difficulty.

**Agreement between SELFEVAL and human evaluation.** We use both human evaluation and SELFEVAL to evaluate the four different diffusion models in Figure 4. Human evaluation performance, measured using pairwise comparison, follows the same ranking as given by SELFEVAL when comparing both types of pixel diffusion models and both types of latent diffusion models. To the best of our knowledge, ours is the first work to establish correlation between the discriminative performance of generative models and human evaluation for text-to-image diffusion models across a wide range of models and tasks. The high degree of alignment between SELFEVAL and human evaluation suggests that SELFEVAL is a reliable and interpretable way to evaluate and compare the text faithfulness of different diffusion models.

Next, we use SELFEVAL to further analyze the performance of diffusion models.

**Effect of the text encoder.** Comparing the different text-encoders used in Figure 4, we observe that diffusion models using the stronger T5 text encoder perform better on most tasks than the ones using the CLIP text encoder. The stronger performance of T5-based models holds for both human evaluations and SELFEVAL. The SELFEVAL results also show that diffusion models using the CLIP based encoders have poor performance, worse than random chance, on the `Count` task. On the `Text Corruption` task that involves identifying a linguistically correct sentence amongst distractors with a shuffled word order, CLIP-based models show a lower performance. Thus, similar to prior work Yuksekgonul et al. (2023), CLIP models show a bag-of-words understanding of the input text and are less sensitive to word order.

**Pixel vs. latent diffusion.** We compare the SELFEVAL performance of the pixel diffusion models to that of the latent diffusion models in Figure 5. Among models that use the same text encoder, *i.e.* PDM-T5 and LDM-T5, we observe that the latent diffusion models outperform the pixel diffusion ones in most cases, especially on the harder tasks of `Attribute Binding`, `Count`, `Spatial Relations` and `Text Corruption`. We hypothesize that this difference can be explained by the fact that the latent diffusion models operate on the compressed latent space and prioritize the text conditioning while 'offloading' the high-frequency image details

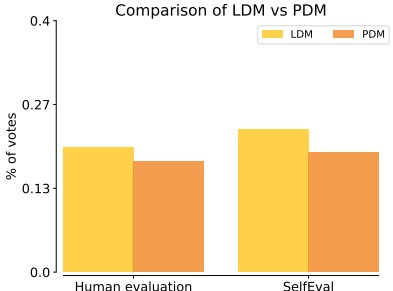

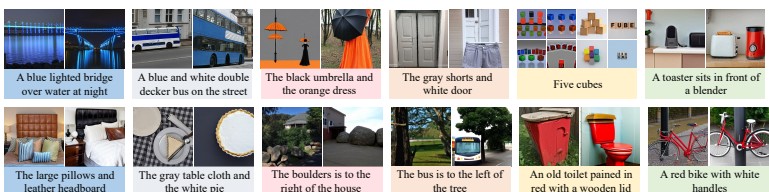

**Figure 6: Qualitative Results.** (Top): Each example compares the generations of pixel diffusion models with CLIP (left) and T5 (right) text encoders. As the difficulty of the prompt increases, models with stronger text encoders maintain higher text fidelity. Both the models fail on simple prompts from `Count` and `Spatial relationships`. (Bottom): Comparison between generations of Pixel (left) and Latent (right) diffusion models with a T5 text encoder. Latent diffusion models can get smaller details like "gray cloth" and "white handles" (second and last example respectively) correctly.

**Figure 5: Pixel vs Latent diffusion.** We observe that human raters rank the generation of latent models higher than pixel models in text faithfulness. We notice a similar ordering using SELFEVAL.

to the autoencoder. We further investigate the performance of pixel and latent diffusion models by employing human raters to evaluate their text faithfulness in Figure 5. The data, for human evaluation, is constructed by randomly picking 500 examples from the all the tasks (100 examples from each task except text corruption), and choosing the right caption as the text prompt. We convert the accuracy of SELFEVAL, to votes, by counting the number of samples where only one model is right. From Figure 5, we observe that human raters prefer generations of latent diffusion models to pixel diffusion ones for text faithfulness. SELFEVAL also shows that latent diffusion models have a better text faithfulness and shows an alignment with human evaluations.

**Qualitative results.** Figure 6 (Top) compares the generations of pixel diffusion models that use T5 and CLIP text encoders. In each example, the image on the left and right are generated using CLIP and T5 text encoder respectively. We notice that as the difficulty of prompts increases, models with a stronger text encoder performs better. Both the models fail on the much harder task of counting instances and spatial relationships. In Figure 6 (Bottom), each example consists two images generated using a pixel diffusion model (left) and a latent diffusion model (right) with a T5 text encoder. We observe that unlike the pixel diffusion model, the latent diffusion model can get small yet important details right ("gray table cloth" and "white handles" in the second and sixth example respectively). We believe that the latent diffusion model can offload the high frequency appearance details to the autoencoder, allowing it to pay more attention to the conditioning variable.

### 4.3 GENERATIVE MODELS APPLIED TO OTHER REASONING TASKS

We now use the challenging Winoground Thrush et al. (2022) benchmark to evaluate the vision-language reasoning abilities of diffusion models. Winoground defines two tasks - (1) 'text score' that involves choosing the right text prompt amongst distractors given an input image; and (2) 'image score' that involves picking the right image amongst distractor images given a text prompt.

**SELFEVAL *vs*. concurrent work** Concurrent work from Li et al. (2023a) demonstrates that diffusion models perform well on the Winoground text score task and achieve competitive performance with discriminative models. Using their formulation yields poor results (zero accuracy) on the image score task as shown in Table 1. Li et al. (2023a) use the ELBO loss as a proxy for the likelihood $p(\mathbf{x}|\mathbf{c})$ which works well for comparing different text prompts and thus leads to good text score performance. However, our analysis revealed that the ELBO loss computed for the predictions from two different images are not comparable, and thus leads to zero image score. SELFEVAL on the other hand, doesn't approximate the likelihood but instead estimates it as described in Sec 3. Using SELFEVAL leads to a non-zero image-score for the same generative model used by Li et al. (2023a), and yields performance close to that of the discriminative CLIP ViT-L model.

**SELFEVAL applied to other diffusion models.** Using SELFEVAL reveals that all the diffusion models introduced in § 4.1.1 achieve competitive performance on both the image score and text score tasks. Compared to all the discriminative CLIP models, generative models achieve strong results in both image and text scores using SELFEVAL. This result reinforces the notion that optimizing the generative objective can provide non-trivial and complementary improvements for several visuo-linguistic reasoning tasks. For additional analysis on the effect of various hyperparameters on the Winoground performance, refer to the supplement.

**Table 1: Diffusion models evaluated on the Winoground dataset**. We measure the image score (accuracy of picking the correct image given a text prompt) and text score (accuracy of picking the correct text given an image). Using SELFEVAL allows us to use diffusion models for both tasks unlike prior work Li et al. (2023a) which leads to zero image score.

| Method | Model | Image Score | Text Score |
|---|---|---|---|
| CLIP (ViT-L/14) | − | 8.00 | 30.25 |
| OCLIP (ViT-H/14) | − | 12.75 | 30.75 |
| Li et al. (2023a) | LDM-CLIP | 0 | 34.00 |
| SELFEVAL | LDM-CLIP | 7.25 | 22.75 |
| SELFEVAL | PDM-CLIP | 14.00 | 17.00 |
| SELFEVAL | PDM-T5 | 12.00 | 28.25 |
| SELFEVAL | LDM-T5 | 13.50 | 29.00 |

**Table 2: Performance of CLIP on the benchmark.** We evaluate the zero-shot performance of CLIP (ViT-L/14 visual backbone) on the six tasks. "Random" denotes the random chance accuracy. CLIP achieves impressive performance on the tasks of `Color` and `Shape`. We observe that the performance of CLIP is close to random on `Attribute binding`, `Count`, `Spatial` and `Text corruption`. This makes CLIP unsuitable for evaluating text faithfulness of generative models on prompts from these tasks.

| Model | Attribute binding | Color | Count | Shape | Spatial | Text corruption |
|---|---|---|---|---|---|---|
| Random | 50 | 25 | 25 | 33 | 25 | 20 |
| CLIP | 55.40 | 85.20 | 67.80 | 91.10 | 40.50 | 51.00 |

## 4.4 DRAWBACKS OF CLIP SCORE

In this section, we discuss a few limitations of the CLIP score that SELFEVAL can effectively address. CLIP score is the most common metric for evaluating text faithfulness of generative models by measuring the cosine similarity between the features of the generated image and the conditioned text caption.

**Sensitivity to the exact CLIP model.**

We report the CLIP similarity scores of the generations from two versions of the Latent Diffusion Models Rombach et al. (2022) on prompts from DrawBench Saharia et al. (2022), Winoground Thrush et al. (2022) and COCO-minival Lin et al. (2014) datasets in Figure 2. The first model (LDM-CLIP (ViT-L/14)) uses the text encoder of CLIP with ViT-L/14 backbone and the second model (LDM-CLIP (ViT-H/14)) uses the text encoder with OpenCLIP Ilharco et al. (2021) ViT-H/14 visual backbone. Across all the three datasets, we observe that LDM-CLIP (ViT-L/14) ranks higher than LDM-CLIP (ViT-H/14) if a CLIP (ViT-L/14 visual backbone) model is used, but ranks lower with an OpenCLIP (ViT-H/14 visual backbone). Our hypothesis is that images generated by a model using a particular CLIP text encoder may still contain some residual information, which could cause them to receive higher scores when assessed using the same CLIP model. This type of bias was identified by Park et al. (2021) in the context of evaluation of text-to-image models, though not in relation to the CLIP score. We emphasize the need for caution among researchers who employ this metric, particularly concerning this bias. SELFEVAL avoids this problem as we do not employ an external model for evaluation.

**CLIP score is limited by CLIP's performance** and thus using it as a proxy on tasks where CLIP itself has poor performance does not yield meaningful comparsions. While the CLIP model has demonstrated impressive zero-shot performance on several image-text tasks, it has severe limitations on many complex reasoning tasks. We compute the performance of CLIP ViT-L/14 model on the six tasks introduced in § 4.1 and report the results in Table 2. CLIP performs well on `Color` and `Shape` but its performance on all the other tasks is poor. On the widely used DrawBench prompts, 25% of the captions evaluate the generations for attribute binding, counting, spatial relationships and text corruption. Thus, using CLIP to evaluate generations on such prompts in DrawBench is not ideal. SELFEVAL avoids this problem by directly leveraging the diffusion model itself.

## 5 CONCLUSION

In this paper, we presented SELFEVAL which is an automated way to assess the text-understanding capabilities of diffusion models. Since SELFEVAL uses the diffusion model itself to estimate the likelihood of real images given text prompts, it does not require external discriminative models for such assessments. We showed that evaluations using SELFEVAL agree with human evaluations across a range of models and tasks demonstrating that SELFEVAL is a reliable automated metric. We believe that such metrics can greatly speed up research in diffusion models and further research to improve the metrics is necessary. Since SELFEVAL's formulation is not limited to text-to-image diffusion models, we believe it can also serve to evaluate other types of conditioned diffusion models such as text-to-audio, text-to-video *etc*. In the future, we want to further generalize SELFEVAL to work with non-diffusion based generative models.

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
