# SELFEVAL: LEVERAGING THE DISCRIMINATIVE NATURE OF GENERATIVE MODELS FOR EVALUATION

## 1 ADDITIONAL DETAILS OF SELFEVAL

In this section, we provide a detailed algorithm and systematic figure of SELFEVAL in Algorithm 1 and Figure 1 respectively. SELFEVAL iteratively denoises an image, similar to the reverse process of diffusion models, but instead estimates the likelihood of an image-text pair.

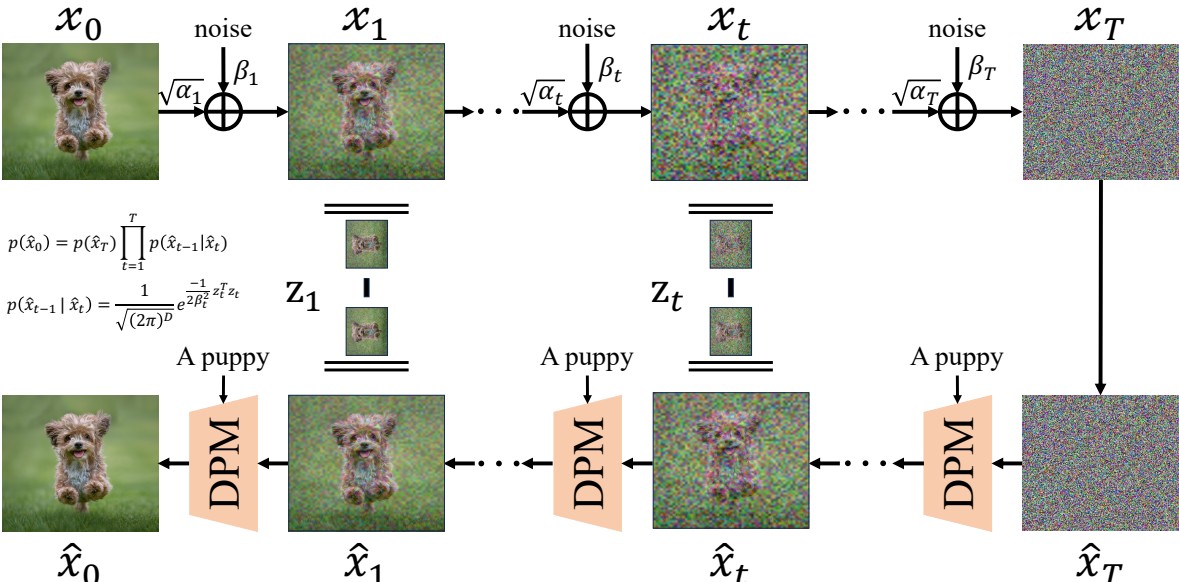

**Figure 1: Illustration of proposed method:** (Left) Starting from a noised input, the standard diffusion sampling method denoises the input iteratively to generate images from the input distribution. (Middle): SelfEval takes an image $x_0$ and conditioning $c$ pairs to estimates the likelihood $p(x_0|c)$ of the pair in an iterative fashion. (Right): Given an image, $x_0$ and $n$ captions, $\{c_0, c_1, \ldots, c_n\}$, SelfEval is a principled way to convert generative models into discriminative models. In this work, we show that the classification performance of these classifiers can be used to evaluate the generative capabilities.

## 2 DETAILS OF HUMAN EVALUATION

Human evaluations are the de-facto standard for judging the performance of text-to-image models. we adopt a conventional A/B testing approach, wherein raters are presented with generations from two models and are asked to vote for one of four choices: "both" the generations are faithful, "none" of them are faithful, or if only one of the two models ("model 1" or "model 2") demonstrates fidelity to the given prompt. We show the template provided to the raters in Figure 2. The template includes three examples that advice the raters on how to rate a given sample followed by a text prompt and two images. The four possible choices are shown on the right in Figure 3. The images used as instructions for the human raters are shown in Figure 3. Figure 3 shows three pairs of images with the text prompt below them. The first example shows two images that are faithful to the input prompt but the quality of one (left) image superior to the other (right). Since, we ask the raters to

---

**Algorithm 1** Algorithm for estimating $p(\mathbf{x}|\mathbf{c})$ using SELFEVAL

---

1: **Input**: Diffusion model $p_\theta(\mathbf{x}_{t-1}|\mathbf{x}_t)$; Input image $\mathbf{x}_0$; Forward latents: $\{\mathbf{x}_{1:T}\}$; Reverse latents: $\{\hat{x}_{1:T}\}$; Number of trials: $N$
2: **for** i=1:$N$ **do**
3:     Sample noise $\sim \mathcal{N}(0, \mathbb{I})$
4:     $\mathbf{x}_{1:T} = q_{\text{sample}}(\mathbf{x}_0, t = 1 : T, \text{noise} = \text{noise}); \mathbf{x}_t \in \mathbb{R}^D$
5:     conditionals $\longleftarrow$ [ ]
6:     **for** j=1:$T$ **do**
7:         $p(\mathbf{x}_{t-1}|\bar{\mathbf{x}}_t, \mathbf{c}) = \frac{1}{\sqrt{(2\pi)^D|\mathbf{\Sigma}_\theta|}} e^{-0.5(\mathbf{x}_{t-1}-\boldsymbol{\mu}_\theta(\bar{\mathbf{x}}_t,t,\mathbf{c}))^T \mathbf{\Sigma}_\theta^{-1}(\mathbf{x}_{t-1}-\boldsymbol{\mu}_\theta(\bar{\mathbf{x}}_t,t,\mathbf{c}))}$
8:         conditionals = [conditionals ; $p(\mathbf{x}_{t-1}|\bar{\mathbf{x}}_t, \mathbf{c})$]
9:     **end for**
10:     Compute $p(\mathbf{x}_T) = \frac{1}{\sqrt{(2\pi)^D}} e^{\frac{-1}{2\beta_T^2}\|\mathbf{x}_T\|^2}$
11:     Compute likelihood $p_i(\mathbf{x}_0|\mathbf{c}) = p(\mathbf{x}_T) \prod_{t=1}^T p(\mathbf{x}_{t-1}|\bar{\mathbf{x}}_t, \mathbf{c})$
12: **end for**
13: $p(\mathbf{c}|\mathbf{x}_0) = \frac{p(\mathbf{x}_0|\mathbf{c})}{|\mathcal{C}|}$

---

evaluate the text faithfulness, we recommend picking the "both" option for such samples. The second image shows an example where only one of the images is faithful to the text. The raters are instructed to pick the option corresponding to the right image in this case. The final example shows two images that are not faithful to the text prompt. The raters are adviced to pick the "none" option in this scenario.

## 3 ABLATION EXPERIMENTS

**Table 1: Effect of timesteps** on the performance of SELFEVAL on the six splits.

| T | Attribute | Color | Count | Shape | Spatial | Text Corruption |
|---|---|---|---|---|---|---|
| 50 | 54.2 | 32.2 | 26.3 | 34.9 | 33.0 | 25 |
| 100 | 54.3 | 34 | 25.8 | 30.2 | 38.0 | 24.3 |
| 250 | 53 | 32.3 | 27.4 | 35 | 32.7 | 21.7 |

**Table 2: Effect of N** on the performance of SELFEVAL on the six splits.

| N | Attribute | Color | Count | Shape | Spatial | Text Corruption |
|---|---|---|---|---|---|---|
| 1 | 53.0 | 26.0 | 27.2 | 35.2 | 31.2 | 20.7 |
| 5 | 54.3 | 31.7 | 25.7 | 34.9 | 33.0 | 22.1 |
| 10 | 54.3 | 34.0 | 25.8 | 32.5 | 38.6 | 24.3 |
| 15 | 53.4 | 36.3 | 28.0 | 36.3 | 32.8 | 22.8 |

**Table 3: Effect of the choice of seed** on the performance of SELFEVAL.

| S | Attribute | Color | Count | Shape | Spatial | Text Corruption |
|---|---|---|---|---|---|---|
| 1 | 54.3 | 34.0 | 25.8 | 32.5 | 38.6 | 24.3 |
| 2 | 53.0 | 26.0 | 27.2 | 35.2 | 31.2 | 20.7 |
| 3 | 54.3 | 31.70 | 25.7 | 34.9 | 33.0 | 22.1 |
| std | 0.5 | 0.5 | 0.9 | 1.4 | 1.5 | 0.8 |

In this section we analyze the effect of various components that affect the performance of SELFEVAL on the six splits introduced in the main paper. We use the LDM-T5 model for all our experiments.

**Effect of T**: SELFEVAL has a time complexity of $\mathcal{O}(NT)$ and Table 1 shows the the effect of timesteps on the performance of SELFEVAL. We observe that SELFEVAL achieves the best result at different timesteps for different datasets. We notice that the performance drops as we increase the timesteps from 100 to 250 in most cases. As the number of timesteps increases, we believe that the fraction of them responsible for text faithfulness decrease, resulting in a drop in performance. We find $T = 100$ to be a good tradeoff for performance and speed and is used for all the experiments on the six data splits in this work.

**Effect of N**: Table 2 shows the results of the effect of number of trials $N$ on the performance of SELFEVAL. We observe that $N = 10$ works best across all the six splits and is the default choice for $N$ unless otherwise mentioned.

**Effect of seeds**: SELFEVAL corrupts an input image using standard gaussian noise in each trial and we analyze the effect of the seed on the performance of SELFEVAL in Table 3. We observe that the performance is stable across all the six splits with a standard deviation within 1 percentage point in most of the cases. We report the seed number instead of the actual value for brevity and use the seed 1 as the default choice for all the experiments.

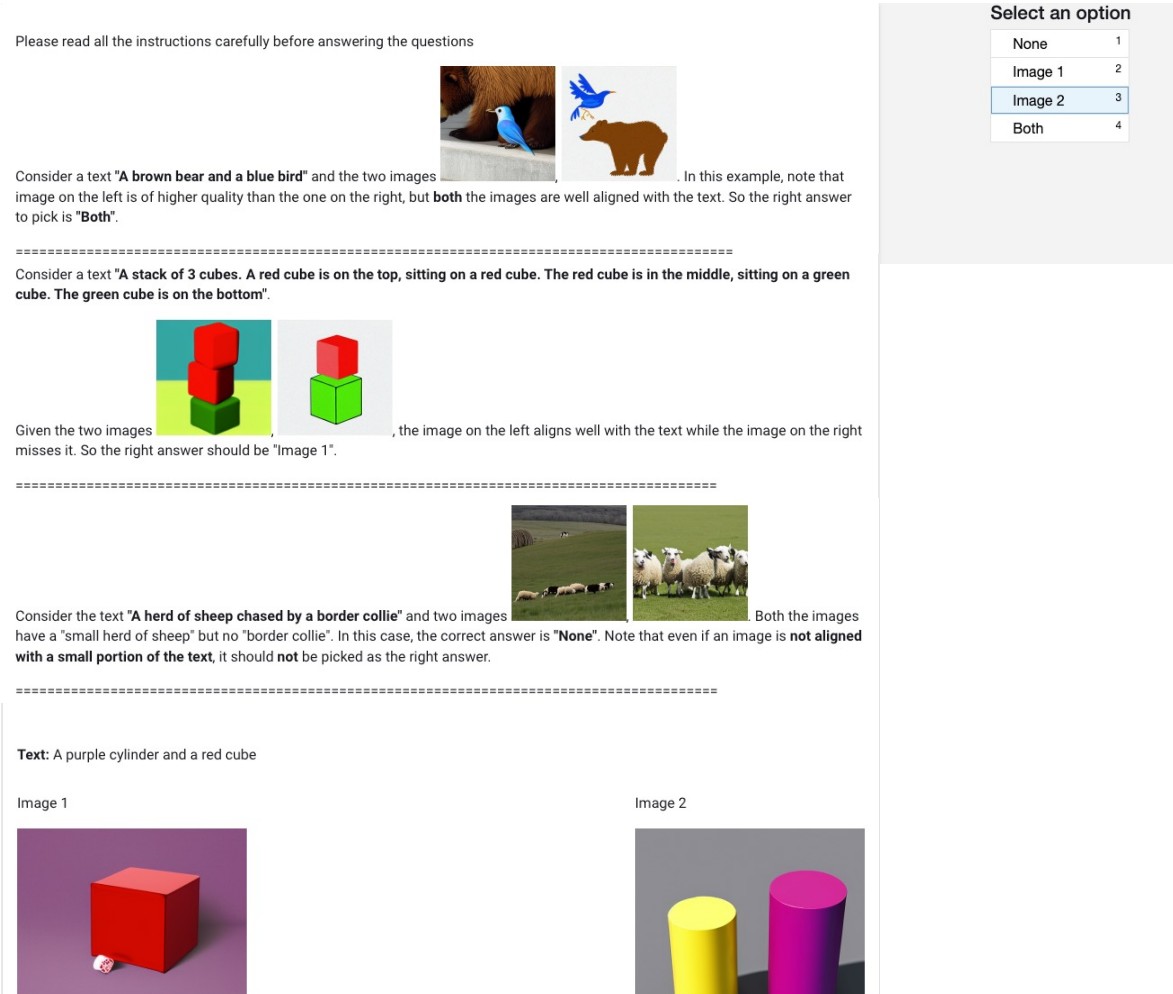

**Figure 2: Template for Human raters.** The template consists of instructions explaining the nature of the task (top) followed by a text prompt with two generations (bottom). Humans are expected to pick one of four options (shown on the right): "both" the generations are faithful, "none" of them are faithful, or if only one of the two images ("Image 1" or "Image 2") demonstrates fidelity to the text prompt.

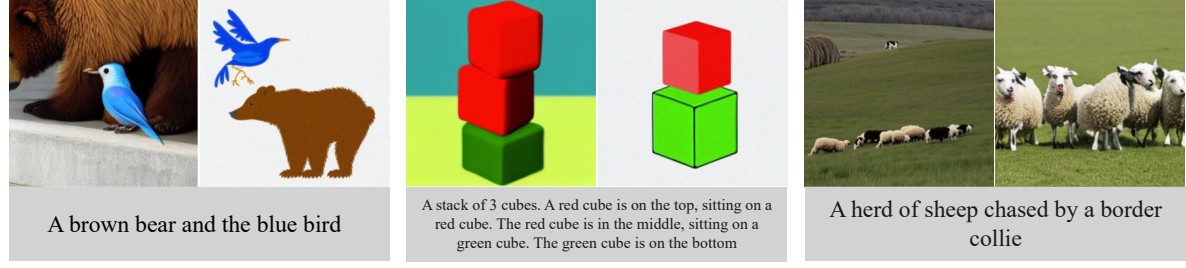

| A brown bear and the blue bird | A stack of 3 cubes. A red cube is on the top, sitting on a red cube. The red cube is in the middle, sitting on a green cube. The green cube is on the bottom | A herd of sheep chased by a border collie |

**Figure 3: Instructions for Human raters.** We provide three examples describing all the possible scenarios. The first example shows two images that are faithful to the text but with varying image qualities. To prevent the raters from conflating image quality with text faithfulness, we recommend the raters to pick "both" for such examples. The second example illustrates a case where only one of the image is faithful to the text. In this case, the raters are adviced to pick the option corresponding to the right image ("Image 1" in this case). The final example shows a case where both the examples are not faithful to the text (there is no border collie), in which case, we advice the raters to pick "none".

## 4 ADDITIONAL EXPERIMENTS ON WINOGROUND

In this section we ablate a few design decisions on the Winoground dataset. We use the LDM-T5 model for all the experiments.

**Table 4: Effect of timesteps** on the performance of SELFEVAL on the Winoground dataset

| T | Image Score | Text Score |
|---|---|---|
| 20 | 11.50 | 30.75 |
| 50 | 13.50 | 29.00 |
| 100 | 12.25 | 25.25 |
| 250 | 11.25 | 27.75 |

**Table 5: Effect of the number of trials** on the performance of SELFEVAL on the Winoground dataset

| N | Image Score | Text Score |
|---|---|---|
| 1 | 17.00 | 26.25 |
| 5 | 14.75 | 26.00 |
| 10 | 13.50 | 29.00 |
| 20 | 11.25 | 24.75 |

**Table 6: Effect of the choice of seed** on the performance of SELFEVAL on the Winoground dataset

| S | Image Score | Text Score |
|---|---|---|
| 1 | 13.50 | 29.00 |
| 2 | 13.00 | 27.00 |
| 3 | 12.00 | 28.50 |
| | $12.83 \pm 0.76$ | $28.17 \pm 1.04$ |

**Effect of T**: We show the effect of the number of timesteps on the performance of SELFEVAL on the Winoground dataset in Table 4. From Table 4, we observe that SELFEVAL achieves the best result for image and text score at different time steps. Image score is a harder task compared to Text score Thrush et al. (2022) and hence SELFEVAL needs more timesteps to perform better on Image score. As the number of timesteps increase, we observe a drop in both Image and Text scores. Studies Li et al. (2023) show that the earlier timesteps generate low frequency information (responsible for text fidelity), while latter ones are responsible for high frequency appearance details. By increasing the number of timesteps, the fraction of timesteps contributing to improving the faithfulness to text (and thereby image and text scores) decreases, resulting in a drop in performance. All other experiments on Winoground use T=50 unless otherwise specified.

**Effect of N**: We show the effect of the number of trials (N) in Table 5. With fewer trials, the estimates are not reliable and larger trials make it computationally expensive. We observe that we attain a good tradeoff for performance and speed with $N = 10$.

**Effect of the seed**: We show the effect of seed on the performance of SELFEVAL in Table 6. We just report the seed number for brevity. We observe that both the scores are relatively stable across different values of seed. We fix seed #1 for all the experiments in this work.

## 5 CONVERTING COCO IMAGE-CAPTION PAIRS FOR ITM

We use image-caption pairs from COCO for the tasks of `Color`, `Count` and `Spatial relationships`. We use the question answering data collected by authors of TIFA Hu et al. (2023) to construct data for our tasks. We pick only samples constructed from COCO for our purpose. Given question answering samples from TIFA, we identify the corresponding image-caption pair from COCO and replace the correct answer in the caption with the multiple choices to form samples for the task of Image-Text Matching.