# OpenReview forum: "SelfEval: Leveraging the discriminative nature of generative models for evaluation"
_ICLR.cc/2024/Conference — Submitted to ICLR 2024_

### Official Review · Reviewer_M8yh · 2023-10-23

**Soundness:** 2 fair
**Presentation:** 3 good
**Contribution:** 2 fair
**Rating:** 5
**Confidence:** 4

**Summary:**

This paper proposed an automatic evaluation method by measuring text-image faithfulness, showing an agreement with human evaluations in multiple benchmarks. By arguing that CLIP-score has severe drawbacks in DrawBench, for example, they argued that their method, SelfEval, solves or detours these issues, hoping for its role as a reliable and easy-access method to evaluate multimodal diffusion models.

**Strengths:**

- They showed the limitation of CLIP-score in quantitative evaluation with extensive analysis.

- One may exploit a strong text-to-image generativel model to evaluate others.

**Weaknesses:**

- W1. Novelty. The motivation of their method is reminiscent of the CLIP-R-Precision of Park et al. (2021), where the goal is to pick the right caption among distractors. Although the authors used the likelihood estimations to get a score for text-image matching, it can be seen as an ad-hoc method to replace the cosine similarity in CLIP. Since we cannot control the pretrained models where one is CLIP and the other one is diffusion models, the authors need to study the effectiveness of the proposed method extensively. The limitation of the CLIP score was also explicitly explored in MID [2].

- W2. State-of-the-art comparison. For the text-image faithfulness, you could compare the state-of-the-art performance with DALL-Eval [1], MID [2], LLMScore [3], and VPEval [4]. The paper failed to cite them. Please allocate the dedicated paragraph in Sec. 2 to include them and compare the related works.

[1] Cho, J. et al. (2022). DALL-Eval: Probing the Reasoning Skills and Social Biases of Text-to-Image Generative Transformers. http://arxiv.org/abs/2202.04053

[2] Kim, J.-H. et al. (2022). Mutual Information Divergence: A Unified Metric for Multimodal Generative Models. Advances in Neural Information Processing Systems 35. http://arxiv.org/abs/2205.13445

[3] Lu, Y. et al. (2023). LLMScore: Unveiling the Power of Large Language Models in Text-to-Image Synthesis Evaluation. http://arxiv.org/abs/2305.11116

[4] Cho, J. et al. (2023). VPGen & VPEval: Visual Programming for Text-to-Image Generation and Evaluation. https://arxiv.org/pdf/2305.15328.pdf

**Questions:**

Could you let me know why you excluded the closely related works mentioned in Weaknesses W2?

---

> ### Author Response · Authors · 2023-11-16
> **Response to Reviewer M8yh**
>
> We thank the reviewer for their valuable time and effort. We resolve some confusions and address your concerns below.
>
> **Clarification about SelfEval**
>
> - We wish to clear the confusion of the reviewer regarding their statement that a stronger text-to-image model can be used to evaluate other models. SelfEval eliminates the need for any external model by converting a generative model ($p(x,y)$) to a discriminative model ($p(y|x)$) and evaluate the generative capabilities using its discriminative performance (proportionality via Bayes’ Rule). We DO NOT use one text-to-image model to evaluate others in this work.
>
> **Weakness**
>
> - We point out some important distinctions of our work from CLIP R-Precision. CLIP R-precision measures the text retrieval performance given a **generated** image whereas in our case, we use the **ground truth** image-text pairs for evaluation. Unlike such embedding-based metrics, our evaluation is absolute, since models are always evaluated against ground truth. This along with the splits introduced in our work, provide a diagnostic tool for analyzing generative models. Another key takeaway and novelty of SelfEval is the fact that a generative model can be used to evaluate itself. To the best of our knowledge, this capability has never been shown before and differentiates our work from other evaluation metrics proposed in literature.
>
> - Limitations of CLIP were highlighted in MID [2] but their analysis is limited and doesn’t show clearly how this affects the final evaluation. In our work, we show the limitations of CLIP’s text understanding and how using different CLIP models affects the actual evaluation.
>
> - We believe [4] was made public AFTER the ICLR paper submission deadline. Except method [2], none of the other works are published in literature. Moreover, methods [1-3] use external models (like CLIP) for evaluation and as mentioned throughout our paper, using external models requires additional error analysis to prevent any biases. With SelfEval, we aim to eliminate any external model for evaluating text faithfulness of generative models and we believe it is unfair to compare SelfEval against these methods. We will add a separate section in the related works to discuss these methods.

---

> > ### Author Response · Authors · 2023-11-18
> > **Follow-up with Reviewer**
> >
> > Dear Reviewer,
> > Kindly let us know if we can further answer your concerns.

---

> ### Comment · Reviewer_M8yh · 2023-11-20
> **Feedbacks**
>
> Thanks for the clarification, but I mean that's a strong point in your work since you repurpose the text-to-image generative model to a discriminative model for evaluating its generative capabilities; if you're targeting the stronger text-to-image generative model ($p(x,y)$), you can have more strong evaluation quality by assuming that its discriminative ($p(y|x)$) performance is stronger.
>
> I acknowledged that the related works [1,3,4] have not been published yet, although LLMScore [3] will be published in NeurIPS 2023, and the ICLR 2024 reviewing guidelines have loose rules for it. Still, you could compare your work with MID [2] along with CLIP, and the failure of this undermined the comparative effectiveness of your work. Nevertheless, I highly recommend faithfully comparing with those related works.

---

> > ### Author Response · Authors · 2023-11-21
> > **Response to Reviewer Feedback**
> >
> > We thank the reviewer for the clarification and we indeed agree with them that as the generative models keep improving, SelfEval gets more robust and reliable.
> > We would like to reiterate that evaluation metrics that use external models like CLIP or LLMs, with their own set of limitations, are unfit for evaluation.
> > Instead, we do not use any external model for evaluations and repurpose the generative model to evaluate its abilities by leveraging the ground truth from image-text recognition datasets.
> > We appreciate the reviewer’s detailed recommendation and evaluate MID on our splits and we observe that 1) Similar to FID, MID requires abundant samples to reliably estimate the scores, unlike our approach (which requires an order of magnitude fewer samples) and 2) MID scores do not correlate well with human judgements.
> >
> > We use the official code released by the authors and follow the exact same data processing steps recommended by the authors. We replace the generated images released by the authors with the generations of our models. In addition to the data used by SelfEval, MID needs more samples for a reliable estimate. Thus, for MID, we use an additional 28,000 samples from COCO2014 validation.
> > The results are as follows. In both the tables, for MID, lower score is better and the performance of SelfEval is the accuracy over chance (similar to main paper). For human evaluations, we report the number of votes received by each method (higher the better).
> >
> > Tab 1:PDM-CLIP vs PDM-T5 (Scores A/B - A: Score of PDM-CLIP, B: Score of PDM-T5)
> > | Method |  Attribute   |     Color    |     Shape    |     Spatial    |
> > |-----------|--------------|--------------|--------------|---------------|
> > | Humans|    24/117    |     29/42     |      30/7     |       19/72    |
> > | MID      |  27.1/33.0  | 38.7/38.6 |  31.6/28.2   |  10.5/29.1   |
> > |SelfEval|    0.4/3.3     |   3.5/5.8    |   0.2/1.3    |     8.6/9.0    |
> >
> > Tab. 2: LDM-CLIP vs LDM-T5 (Scores A/B - A: Score of LDM-CLIP, B: Score of LDM-T5)
> >
> > | Method |  Attribute   |     Color    |     Shape    |      Spatial     |
> > |-----------|--------------|--------------|--------------|-----------------|
> > | Humans|    14/140    |     27/69    |     25/48     |       21/73      |
> > | MID      |  25.1/33.6  | 32.4/40.8  | 17.0/27.7  |   16.8/30.4     |
> > |SelfEval|    1.0/4.1     |  8.0/9.4     |    0.8/-1.1  |     8.4/10.8     |
> >
> > We observe a significant disagreement with human judgements across all splits and families of models for MID. On the easier splits of **Color** and **Shape**, the gap in MID scores for the two methods is lower compared to the harder splits across both the families. We attribute this to the limitation of the CLIP model. Due to the lack of good reasoning abilities on the harder splits, the gap in MID scores is higher, with significant disagreements with human judgements, resulting in low reliability of MID.
> > We will add this discussion to the next version.

---

> ### Comment · Reviewer_M8yh · 2023-11-22
> **Additional feedbacks for the comparison**
>
> I appreciate your endeavor to provide the requested comparison and acknowledge that MID requires a sufficient number of comparative samples like FID.
>
> * I need more elaboration on the details of the experiments.
>   - "In both the tables, for MID, lower score is better" -> for the MID score, the higher wasn't better?
>   - "SelfEval is the accuracy over chance" -> how do you get the accuracy? What brings you to this conclusion?
>   - I only understand that Humans A/B in the tables means that the number of humans voting for the generations from the "A" model is A, and for the generations from "B" is B.
>
> * Do you assume that since B is more voted than A by humans, the evaluation metrics should vote for B? A or B cannot give consistent generation performance for every sample, so you should consider the sample-wise preferences of humans. For this reason, many related works resort to Kendal-tau correlations to see if a model follows human judgments for the samples. Since the reports missed key details and ambiguity, my interpretations are limited. Please correct me if there's a misunderstanding.

---

> > ### Author Response · Authors · 2023-11-22
> > **Response to Reviewer M8yh**
> >
> > We thank the reviewer for engaging with us and helping us improve our work.
> >
> > For a set measuring a particular quantity (say color), if model B gets more votes from humans than model A, we observe that accuracy of model B is higher than model A on a classification task measuring that quantity.
> >
> > We agree with the reviewer that many related works resort to Likert scale type ratings. However, there are works like [*] that conclude that Likert scale is not necessarily better than pairwise comparison in human evaluations for several reasons. For text faithfulness, which is a highly subjective task, it is very difficult to define a rubric (which score to assign for a small mistake in the generation) for evaluation, especially for **untrained** human evaluators making the results extremely biased. On the other hand, pairwise evaluation is a more stricter evaluation while simultaneously making the rubric much simpler (we encourage the raters to reject a generation if there is any mistake big or small).
> >
> > We address your questions below
> > - We thank the reviewer for pointing it out. There is quite a bit of confusion around this in our opinion. The metric is called Mutual Information **Divergence** and some excerpts from the paper, like “measuring the divergence from the ground-truth or reference samples” suggest that MID measures distance. From statistical theory, divergence measures the difference between two distributions and that is the reason we interpret it as “small is better”. However, after looking into the paper in detail, the reviewer is correct and higher numbers are better. In spite of this, we still observe the issue with CLIP as seen in Fig. 2 (left) in the main paper. We repeated that experiment on Winoground images, and used MID (instead of CLIP score) to compute the scores using CLIP ViT-B/32 and ViT-L/14 backbones. The results are shown in the table below. With ViT-B/32 backbone for evaluation, LDM-CLIP (ViT-H/14) ranks better than LDM-CLIP (ViT-L/14) whereas with ViT-L/14 backbone, LDM-CLIP (ViT-L/14) ranks better than LDM-CLIP (ViT-H/14). This reinforces the point that the biases and limitations of the underlying external model significantly affect the performance of the metric. This can be effectively avoided by SelfEval. We will add this discussion to the main paper.
> >
> > |             Model            |  ViT-B/32 | ViT-L/14  |
> > |-------------------------|------------|------------|
> > | LDM-CLIP (ViT-L/14) |    27.77    |     25.70   |
> > | LDM-CLIP (ViT-H/14) |    29.25    |    23.53    |
> >
> > - As mentioned in the paper, we pose the evaluation as accuracy of picking the right caption among distractors. The number of distractors are different for each split, stipulated by the creators of the dataset. Hence, we decided to report accuracy over chance to avoid difficulties in presentation. For example, if chance accuracy for a classification task is 25% and a model achieves 28%, then we report 3%. Note that this is just for ease and not a requirement.
> >
> > - Yes, that is correct.
> >
> >
> > [*] Tyumeneva Y, Sudorgina Y, Kislyonkova A, Lebedeva M. Ordering motivation and Likert scale ratings: When a numeric scale is not necessarily better. Front Psychol. 2022 Sep 23.

---

> ### Comment · Reviewer_M8yh · 2023-11-23
> **Feedbacks to the response**
>
> * Your argument that Likert scale-type ratings can be subject to bias is reasonable, although a sufficient number of samples would show better human judgment correlations for the better model.
>
> * I agree with the authors that the proposed model has advantages over previous works, including MID, since the evaluation is not biased from the other pretrained models when there's no significant bias in the generative model itself.
>
> * Unfortunately, I cannot see that the proposed method outperforms or is on par with MID. In the above Tbl 1 & 2, if I'm correctly interpreting, both MID and SelfEval roughly follow human voting tendencies. However, since MID and SelfEval's metrics are different, the MID score and the Accuracy Over Chance, respectively, we cannot compare directly. Couldn't you convert MID scores to the Accuracy Over Chance compatibly? Please don't mix up different metrics in one table. Because the authors argued that MID is subject to the pretrained model's bias, you can *easily* show that SelfEval may outperform MID, but the current experiment cannot show it, making the work incomplete.
>
> * After discussion, I cannot help to raise my score but remain leaning toward weak rejection due to the above concern.

---

> > ### Author Response · Authors · 2023-11-23
> > **Response to Reviewer**
> >
> > We thank the reviewer for agreeing with the advantages of SelfEval over previous works, including MID, our reasoning supporting pairwise evaluations and for improving their score after the discussion.
> >
> > - We believe the reviewer has raised another important shortcoming of metrics that compute some form of similarity scores (CLIP-Score, MID etc.). Unlike accuracy (used by SelfEval), such scores are not interpretable making it challenging to promptly assess the quality of the models (how good are models with a particular score). SelfEval computes accuracy using ground truth samples, providing an interpretable and clear indication of the models' performance (models with a particular accuracy). Unfortunately, there is no straightforward way to convert the MID score to Accuracy and we believe that is a shortcoming of MID and not SelfEval.
> > - As previously mentioned, there is a lot of confusion regarding the name and language used in MID as it doesn’t measure divergence making the results mildly questionable. We request the reviewer to provide additional details; since they consider our work to be incomplete, despite agreeing with the advantages of our work and rightfully recognizing that SelfEval is complementary to existing works.
> > - We point out that such biases keep affecting metrics as long as an external model is used for evaluations and research such as SelfEval (which is complementary to existing work) is warranted.
> >
> > In light of this discussion we request the reviewer to reconsider their decision.

---

> ### Comment · Reviewer_M8yh · 2023-11-23
>
> In Tbl. 3 of the MID paper, they compared in accuracy. In a similar way, you could get the accuracy using MID.
>
> As Sec. 3.3 in this manuscript stated that you measured the accuracy by picking the correct caption for a single image $x$ from a set of captions $\\{c_i\\}$, you could measure the MID scores between the single image $x$ and each one of the captions $\\{c_i\\}$. Then, you can pick the highest MID score among them as the prediction and measure the accuracy. Notice that MID can give you a pair-wise score like the CLIP-score.
>
> In this way, CLIP and MID scores can be converted to the accuracy having the interpretability you argued.
>
> Since the accuracy metric is based on a set of probing captions, it could be sensitive toward the choice of candidate captions inevitably. Notice that the accuracy also has weaknesses.

---

### Official Review · Reviewer_jCGn · 2023-10-25

**Soundness:** 3 good
**Presentation:** 2 fair
**Contribution:** 2 fair
**Rating:** 6
**Confidence:** 3

**Summary:**

This paper proposes a method that can evaluate text-faithfulness of text-to-image generative models. The main idea is to employ the generative model itself for discriminative tasks, thereby evaluating the performance. Experimental results show that the proposed method can achieve a high degree of agreement for measuring text-faithfulness with human evaluations on several models and datasets, proving the effectiveness.

**Strengths:**

1. The method is simple but effective, which can be easily implemented and applied to other tasks.
2. Experiments show the effectiveness and good performance for measuring text-faithfulness.

**Weaknesses:**

As the authors said, the formulation can extend to any conditional diffusion model. However, the current analysis and experiments only demonstrate that the method is effective for text-to-image generation tasks with diffusion models.

1. Considering the method only evaluates the text-faithfulness without image quality in text-to-image generation tasks, the application scope will be very limited. Therefore, the contribution is not overwhelming and might not be enough for this conference, in my opinion.

2. The experiments are also limited. Since the proposed method is an evaluation metric, trying it to evaluate other frameworks, such as VAEs and GANs, is also necessary. If the method can only be applied to diffusion models, it will be less valuable.

**Questions:**

Although diffusion models have become very popular recently, this framework has not become the only generative model. Meanwhile, other frameworks also perform well in text-to-image generation tasks. Therefore, I think the method has a limited application scope. In my opinion, the authors should try other conditional generation tasks rather than only text-to-image generation tasks, or try applying the method to other generative models.

However, in addition to this problem, I cannot find other technical flaws or missed experiments. Thus, I tend to a marginal score.

---

> ### Author Response · Authors · 2023-11-16
> **Response to Reviewer jCGn**
>
> We thank the reviewer for recognizing the simplicity and effectiveness of SelfEval for evaluating text faithfulness of generative models. Please find our responses to your concerns and questions below.
>
> **Weaknesses**
>
> - Using Eq. 4-9 (in the main paper)  one can compute the likelihood of the data $x$ given conditioning $c$.  Throughout the derivation we do not make any assumptions on the shape/format of the conditioning variable $c$. We chose to proceed with text-to-image generation because evaluating the performance of the model on the image-text matching task is correlated with its generation capabilities. If the reviewer has specific conditioning variables or downstream tasks in mind, please let us know and we can provide a specific and detailed answer.
>
> - We believe that proper evaluation metrics are necessary to make progress in any field. Given the exponential progress in generative modeling, there is a need for developing strong evaluation metrics. We argue that research on evaluation metrics is not narrow but rather necessary to propel the advancement of generative models. Moreover, [1]  proposed CLIP score, widely used for evaluating text faithfulness while [2] proposed FVD a metric to measure quality of generated videos, both of which are widely used by the generative community making such research very valuable.
>
> - We re-iterate that the contribution of our work is broadly two fold. 1) Generative models can themselves be used to evaluate their generations and 2) a principled approach to achieving this with text-to-image diffusion models and repurposing discriminative ground truth image-text pairs. We believe that 1) is an important contribution with far reaching consequences and is not specific to diffusion models. Generative models can inherently be used to evaluate themselves using Bayes’ Rule and this applies to any future developments in this area. We agree with the reviewer that showing results on additional models is beneficial but after an extensive review we fail to obtain any text-to-image GAN or VAE models which are publicly available making it impossible to evaluate using this approach (see common response above). Instead we chose four popular models, with enough details in their respective papers for re-implementation, from two families of diffusion models to validate our hypothesis. We urge the reviewer to provide specific details (papers, codes, checkpoints) of good text-to-image generative models we can include in our analysis.
>
> [1] Jack Hessel et al. CLIPScore: A Reference-free Evaluation Metric for Image Captioning. ”EMNLP (2021)”.
>
> [2] Unterthiner, Thomas et al. “FVD: A new Metric for Video Generation.” DGS@ICLR (2019)

---

> > ### Author Response · Authors · 2023-11-23
> > **Follow-up with Reviewer jCGn**
> >
> > Dear Reviewer,
> > We thank you once again for your time and request you to take a look at the rebuttal posted for your reviews.
> >
> > Additionally, we perform experiments comparing SelfEval with MID[1] as suggested by Reviewer M8yh. MID uses a CLIP model to compute mutual information divergence and we observe that it suffers from the same drawbacks as CLIP-Score (Fig. 2 in the main paper). We show the results in Table-1 below. The second and third column in Table-1 compute MID using ViT-B/32 and ViT-L/14 backbones respectively. Similar to CLIP score, depending on the evaluation model used, LDM-CLIP (with ViT-L/14 text encoder) ranks higher/lower than LDM-CLIP (with ViT-H/14 text encoder). We believe this issue exists regardless of the metric as long as it relies on external models. For more details refer to the review posted to Reviewer M8yh.
> >
> > Table-1: Drawback of MID
> > |             Model               |  ViT-B/32 | ViT-L/14  |
> > |-----------------------------|-------------|-------------|
> > | LDM-CLIP (ViT-L/14) |    27.77    |     25.70   |
> > | LDM-CLIP (ViT-H/14) |    29.25    |    23.53    |
> >
> > Please take a look at our posted rebuttal, additional discussions and provide us with more details. We believe this discussion can convince the reviewer to improve their score.
> >
> > [1] Kim, J.-H. et al. (2022). Mutual Information Divergence: A Unified Metric for Multimodal Generative Models. Advances in Neural Information Processing Systems 35.

---

### Official Review · Reviewer_VHYX · 2023-10-31

**Soundness:** 3 good
**Presentation:** 3 good
**Contribution:** 2 fair
**Rating:** 6
**Confidence:** 3

**Summary:**

This paper proposes a method for evaluating text-to-image generative models using the model's own capabilities. The proposed SELFEVAL approach computes the likelihood of real images given text prompts, enabling the generative model to perform discriminative tasks. This allows for fine-grained evaluation of the model's performance in attribute binding, color recognition, counting, shape recognition, and spatial understanding. Paper claims that SELFEVAL is the first automated metric to show a high degree of agreement for measuring text-faithfulness with human evaluations across multiple models and benchmarks. Unlike other metrics such as CLIP-score, which can show severe drawbacks when measuring text faithfulness, SELFEVAL offers reliable, automated evaluation for diffusion models without additional pre-trained models.

**Strengths:**

1. The proposed method provides a way to automate the evaluation process, making it less dependent on often subjective and time-consuming human evaluations. It allows for detailed assessment of various aspects of a model's capabilities, such as attribute binding and color recognition. The ability to evaluate models without needing additional pre-trained models is a significant advantage.
2. Sufficient evaluations of the proposed metric along with others are given. It shows consistency with human evaluations.

**Weaknesses:**

1. The performance of SELFEVAL is inherently linked to the effectiveness of the generative model itself, which could limit its reliability if the generative model has weaknesses in certain areas. From this perspective, more generative models should be evaluated.
2. The authors don't discuss the potential limitations of their proposed method in detail.

Minor issues:
1) There may exist some misuse between \cite and \citep as some citation formats seem improper in the paper.

**Questions:**

1. How well does the SELFEVAL method generalize across different types of generative models and datasets?
2. Can this method be employed in GAN, VAE, and flow-based generative models?
3. Including a broader range of comparative studies with current methods for assessing the quality of text-to-image generative models could strengthen your arguments.
4. How different text encoders impact the performance of SELFEVAL.

---

> ### Author Response · Authors · 2023-11-16
> **Response to Reviewer VHXY**
>
> We thank the reviewer for recognizing SelfEval’s ability to eliminate the use of external models and its strong agreement with human evaluations.  We will address your concerns and questions below.
>
> **Weaknesses**
>
> - We agree with the reviewer that the limitations of the evaluating model affects the performance of the evaluation itself but as repeatedly mentioned throughout the paper, this is an issue only when an external model is used. SelfEval uses the generative model itself for evaluation. For example, if the generative model is weak in “counting”, then as shown in the paper, the scores for this task are lower. We show that this is the case with four popular text-to-image models and request the reviewer to provide the details (paper, code and checkpoints) of other generative models they deem useful.
>
> - SelfEval relies on the sampling of the generative model to compute the scores. So the limitations of the sampling process of a generative model affect SelfEval. Unlike other black-box evaluation methods which only require the generations from the model, SelfEval requires the model definition and its checkpoints for evaluation. We can provide more details if necessary and will add this discussion to the next draft.
>
> **Questions**
>
> - SelfEval is evaluated using two different families of diffusion models, pixel and latent and across 4 different datasets and shows strong agreement with human evaluators.
> - Any likelihood-based generative model (i.e. ones that model the data distribution $p_{\text{data}}(x)$ , like VAEs) can be used with SelfEval. For implicit generative models, like GANs, methods like GAN inversion [1] can be used to obtain likelihood scores from them as described in [2]. After an extensive survey of existing methods, we failed to obtain public repositories of  text-to-image GANs or VAEs with checkpoints. We request the reviewer to point us to relevant text-to-image generative models (With code and checkpoint) for further experiments.
> - We request the reviewer to clarify what additional analyses/experiments they consider valuable to this work.
> - We show the effect of different text encoders in Sec. 4.2 of the main paper.
>
> [1] W. Xia, Y. Zhang, Y. Yang, J. -H. Xue, B. Zhou and M. -H. Yang, "GAN Inversion: A Survey," in IEEE T- PAMI, 2023.
>
> [2] Eghbal-zadeh, Hamid and Gerhard Widmer. “Likelihood Estimation for Generative Adversarial Networks.” 2017.

---

> > ### Author Response · Authors · 2023-11-18
> > **Follow-up with Reviewer VHXY**
> >
> > Dear Reviewer,
> > Kindly let us know if we can answer any more of your concerns.

---

> ### Author Response · Authors · 2023-11-23
> **Follow-up with Reviewer**
>
> Dear Reviewer,
> We perform additional experiments comparing SelfEval with MID[1] as suggested by Reviewer M8yh. MID uses a CLIP model to compute mutual information divergence and we observe that it suffers from the same drawbacks as CLIP-Score (Fig. 2 in the main paper). We show the results in Table-1 below. The second and third column in Table-1 compute MID using ViT-B/32 and ViT-L/14 backbones respectively. Similar to CLIP score, depending on the evaluation model used, LDM-CLIP (with ViT-L/14 text encoder) ranks higher/lower than LDM-CLIP (with ViT-H/14 text encoder). We believe this issue exists regardless of the metric as long as it relies on external models. For more details refer to the review posted to Reviewer M8yh.
>
> Table-1: Drawback of MID
> |             Model               |  ViT-B/32 | ViT-L/14  |
> |-----------------------------|-------------|-------------|
> | LDM-CLIP (ViT-L/14) |    27.77    |     25.70   |
> | LDM-CLIP (ViT-H/14) |    29.25    |    23.53    |
>
> Please take a look at our posted rebuttal and additional discussions and provide us with more details. We believe this discussion can convince the reviewer to improve their score.
>
> [1] Kim, J.-H. et al. (2022). Mutual Information Divergence: A Unified Metric for Multimodal Generative Models. Advances in Neural Information Processing Systems 35.

---

### Author Response · Authors · 2023-11-16
**Author response to the reviewers.**

We summarize our contributions and clarify some confusions below.

**Contributions**
- We identify several limitations of automatic evaluation methods that use an external model, like CLIP, to measure text-faithfulness of generative models.
- We propose an automatic evaluation for generative models that does not use **ANY** external models but instead repurposes discriminative datasets with **GROUND TRUTH** image-text pairs for evaluating text faithfulness of generative models. So our evaluation 1) sidesteps the issues with CLIP-based or other external model-based evaluations and 2) more reliable and robust since its measured against the ground truth.
- We convert a generative model into a discriminative one and use the discriminative performance to assess its image-text understanding capabilities, eliminating the use of an external model.
- We show a high degree of alignment of this metric with human judgment.

**Response to comments**

- We **DO NOT** use one generative model to evaluate others but instead use it to evaluate itself on text faithfulness. This eliminates the use of an external model for evaluation.
- We show through our analysis that external models, with their own set of limitations, are unfit for evaluating the generative capabilities of a model and we believe that comparing against such methods is not suitable for our setup.
- After an extensive literature review and public benchmarks[*] , we concluded that diffusion models are the most dominant class of generative models with impressive text-to-image generation abilities. We have identified a few models (listed below) that use other classes of generative models but they are not publicly available making it impossible to evaluate them using SelfEval.

**T2I models**

- **GANs**
1. https://mingukkang.github.io/GigaGAN/ -> Not public

2. https://github.com/autonomousvision/stylegan-t -> Not public

- **VAEs** -> No T2I models

- **Token-based Models**
1. Parti - https://sites.research.google/parti/ -> Not public

2. MUSE - https://muse-model.github.io/ -> Not public

- **Class conditioned models**
1. VQ-VAE-2 -> no official code

2. Style-GAN 2 : https://github.com/NVlabs/stylegan2-ada-pytorch (Has only cifar-10 checkpoint.)

[*] https://paperswithcode.com/task/text-to-image-generation

We urge the reviewers to provide specific details of the models and analysis they recommend to help us improve the paper.

---

### Meta-Review · Area_Chair_1szx · 2023-12-06

**Metareview:**

In this paper, authors proposed a method SELFEVAL that uses the generative model to compute the likelihood of real images given text prompts, making the generative model applicable to discriminative tasks, i.e., evaluate generative models in terms of attribute binding, color recognition, counting, shape recognition, spatial understanding, etc. As mentioned by reviewers the strengths of this papers are: 1) a new way to automate the diffusion model evaluation process; 2) method is simple but effective; 3) experiments shows the effectiveness of the methods. Weaknesses are: 1) The performance of SELFEVAL is inherently linked to the effectiveness of the generative model itself, which could limit its reliability if the generative model has weaknesses in certain areas; 2)  They could not use one generative model to evaluate others which makes the application scope limited; 3) experiments are limited.

This paper received 2 "6: marginally above the acceptance threshold" and 1 "5: marginally below the acceptance threshold". But at least one of reviewers who gave 6 rating is not supportive read from comments.

**Justification For Why Not Higher Score:**

Both application scope and experiments are limited.

**Justification For Why Not Lower Score:**

NA

---

### Decision · Program_Chairs · 2024-01-16

Reject